# Virtual 2D map of cyanobacterial proteomes

**Tapan Kumar Mohanta**[1]*, **Yugal Kishore Mohanta**[2], **Satya Kumar Avula**[1], **Amilia Nongbet**[3], **Ahmed Al-Harrasi**[1]*

**1** Natural and Medical Sciences Research Center, University of Nizwa, Nizwa, Oman, **2** Department Applied Biology, University of Science and Technology Meghalaya, Baridua, Meghalaya, India, **3** Department of Botany, University of Science and Technology Meghalaya, Baridua, Meghalaya, India

* nostoc.tapan@gmail.com, tapan.mohanta@unizwa.edu.om (TKM); aharrasi@unizwa.edu.om (AA-H)

**Data Availability Statement:** All relevant data are within the paper and its Supporting Information files.

**Funding:** This study was supported by the Research Council, Oman in the form of a research grant (BFP/RGP/EBR/21/005) awarded to TKM. No

## Abstract

Cyanobacteria are prokaryotic Gram-negative organisms prevalent in nearly all habitats. A detailed proteomics study of Cyanobacteria has not been conducted despite extensive study of their genome sequences. Therefore, we conducted a proteome-wide analysis of the Cyanobacteria proteome and found *Calothrix desertica* as the largest (680331.825 kDa) and *Candidatus synechococcus spongiarum* as the smallest (42726.77 kDa) proteome of the cyanobacterial kingdom. A Cyanobacterial proteome encodes 312.018 amino acids per protein, with a molecular weight of 182173.1324 kDa per proteome. The isoelectric point (*pI*) of the Cyanobacterial proteome ranges from 2.13 to 13.32. It was found that the Cyanobacterial proteome encodes a greater number of acidic-*pI* proteins, and their average *pI* is 6.437. The proteins with higher *pI* are likely to contain repetitive amino acids. A virtual 2D map of Cyanobacterial proteome showed a bimodal distribution of molecular weight and *pI*. Several proteins within the Cyanobacterial proteome were found to encode Selenocysteine (Sec) amino acid, while Pyrrolysine amino acids were not detected. The study can enable us to generate a high-resolution cell map to monitor proteomic dynamics. Through this computational analysis, we can gain a better understanding of the bias in codon usage by analyzing the amino acid composition of the Cyanobacterial proteome.

## Introduction

Cyanobacteria are a diverse group of gram-negative, oxygenic, photosynthetic organisms that originated approximately 2–3 billion years ago [1–4]. Cyanobacteria can inhabit almost any environment, including arid and semi-arid environments and the Arctic and Antarctic [5–8]. They can also tolerate extremely adverse conditions, such as high salinity, low *pH*, and high and low light irradiance [9–12]. They also have the ability to fix atmospheric nitrogen through the use of nitrogenase enzymes [13,14]. Genomic technologies have provided the capacity to unlock the complex interactions between organisms and entire biological systems. The use of genomic and proteomic data has been increasingly utilized in research studies. To provide additional information that are not obtainable with only genomic and transcriptomics-based evaluations, proteomics is a field of study that has received great interest because while the genome of an organism is constant, the proteome differs from cell to cell and undergoes

additional external funding was received for this study. The funders had no role in study design, data collection and analysis, decision to publish, or preparation of the manuscript.

**Competing interests:** The authors have declared that no competing interests exist.

constant change during development and in response to external stimuli [15–17]. The proteome provides more information than the genome, including understanding cellular function and the potential links between gene expression and translation [18–20]. Proteomic technologies enable the exploration of the structure and function of a protein and serve as a connecting link between transcriptomics and metabolomics [21–23]. Therefore, characterizing the details of the proteins of an organism and proteins at a kingdom level is of enormous importance. A comparative proteomic study can provide fundamental information on the role of proteins in complex biological processes, including growth, development, stress response, molecular signaling, etc. The ability to acquire this information requires specific methodologies, including high-throughput technologies, that enable one to identify and isolate a particular protein. Two-dimensional (2D) gel electrophoresis is one of the most prevalent techniques used to separate proteins based on molecular weight and isoelectric points (*pI*) at the *pH* range of 3–11. As a result, we lose a lot of information about the whole proteome study because 2D gel electrophoresis is not possible below *pH* 3 and above *pH* 11. No IPG (immobilized *pH* gradient) stripe exists below *pH* 3 and above *pH* 11. Therefore, it was essential to understand the proteome's detailed *pH* gradient range and provide necessary information about the protein below *pH* 3 and above *pH* 11. Although proteins can undergo post-translational modifications, which result in changes in charge of the protein, the original (without any modification) charge of a protein is still of enormous importance. Since several proteins can undergo reversible post-translational modifications, they can retain their original charge after the modification has been accomplished. Understanding the native molecular weight and isoelectric point of a protein can help to understand its possible function and sub-cellular localization as well.

The chloroplast of algae, plants, and protists are descended from internalized cyanobacterium that retained many cyanobacterial genes with conserved photosynthetic activity. Therefore, knowledge of the proteomic data of the cyanobacterial kingdom will enable us to understand the evolution and biochemical process of its cyanobacterial ancestors. Several studies have been conducted in cyanobacteria that reported the proteome profiles of isolated cellular fractions [24–27]. The studies were conducted in several isolated fractions, including plasma membrane [26], thylakoid membrane [28], outer membrane [29], and soluble fractions [30]. However, there were numerous inconsistencies in protein localization to the sub-cellular fractions [26,31–33]. It was essential to understand the proteomic details of the cyanobacterial kingdom under a single umbrella. Therefore, in the present study, we attempted to determine the molecular weight and isoelectric point of cyanobacterial proteins by considering the annotated (ORF) protein sequences of cyanobacterial proteome and constructed a virtual 2D proteome map using the *in-silico* approach. It also provided important information on the amino acid composition of proteins in cyanobacterial species that can be used to determine codon usage in these organisms.

## Results

### Cyanobacterial proteome ranged from 42726.766 to 680331.825 kDa in size

The molecular weight of the Cyanobacterial proteome was found to range from 42726.766 to 680331.825 kDa. *Candidatus synechococcus spongiarum* (order Melainabacteria) was found to encode the smallest proteome (42726.766 kDa), while *Calothrix desertica* (order Nostocales) encoded the largest proteome (680331.825 kDa) (S1 Table). Other species found to encode smaller proteomes included *Prochlorococcus marinus* (46474.77806 kDa), *Richelia* sp. (48930.82514 kDa), and *Trichodesmium thiebautii* (50144.32502 kDa) (S1 Table). In contrast, species found to contain larger proteomes included *Mastigocoleus testarum* (603575.5146 kDa), *Nostoc punctiforme*, *Oscillatoria acuminata* (457197.2525 kDa), and several others (S1

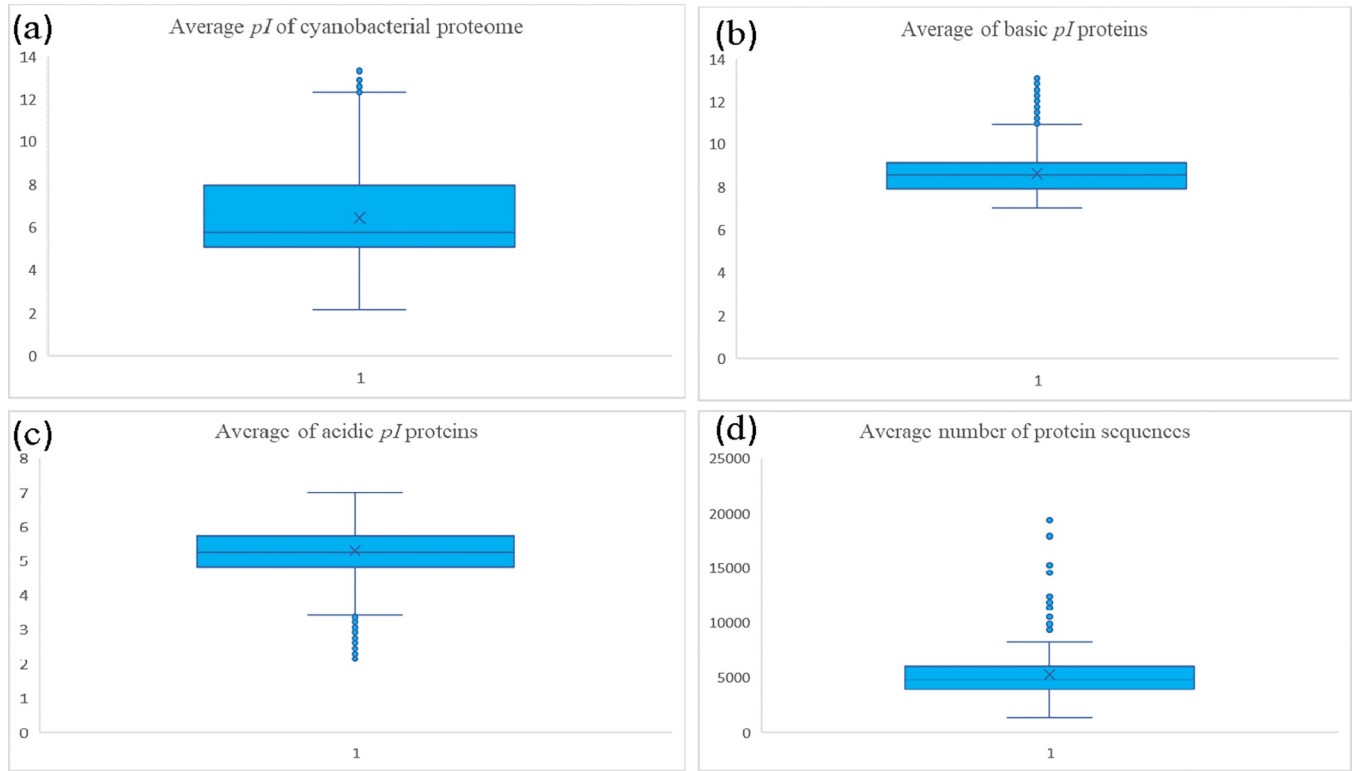

**Fig 1.** Box and whisker plot of (a) average *pI* of cyanobacterial proteins (b) average of basic *pI* proteins (c) average of acidic *pI* proteins and (d) number of protein sequences of cyanobacterial proteomes used in this study. The graphs were prepared using Microsoft excel version 2016.

Table). The average molecular weight of the cyanobacterial proteome was 182173.1324 kDa. *Calothrix desertica* was also encoded the highest number of protein sequences (19335), while *Candidatus synechococcus spongiarum* encoded the lowest (1337). Overall, cyanobacterial species encoded an average of 5260.704 protein sequences (Fig 1). The species, *Anabaena cylindrica*, was found to only encode 17878 protein sequences, and the molecular weight of the proteome of this species was 207570.656 kDa. These data indicate that the number of protein sequences in the proteome is not directly proportional to the molecular weight of the proteome. The proteome of the cyanobacterial species from order Spirulinales is comparatively heavier than the proteome of others (S1 Table). The average molecular weight of the protein of different classes of cyanobacteria increasing order was unclassified (30.519 kDa), Synechococcales (33.063 kDa), Chroococcidiopsidales (34.311 kDa), Chroococcales (34.487 kDa), Gloeobacterales (34.72 kDa), Pleurocapsales (35.571 kDa), Nostocales (35.788), Melainabacteria (36.659 kDa), Oscillatoriales (36.701 kDa), and Spirulinales (37.151 kDa) (S1 File). The molecular weight of proteins from Chroococcidiopsidales, Chroococcales, and Gloeobacterales falls around 34 kDa, whereas the molecular weight of proteins from other groups is slightly different. To further corroborate this premise, a correlation regression analysis was conducted based on the number of protein sequences and the molecular weight of the proteome. Results indicated that the number of protein sequences of the cyanobacterial proteome is only slightly proportional to the molecular weight of the proteome (kDa) (S1 Fig). Although the correlation was positive ($r = 0.918$, $y = 13570 + 32.07 x$), it was $< 1$ (S1 Fig). The D'Agostino and Pearson omnibus normality test indicated that the cyanobacterial proteome did not come from a normally distributed population (it did not pass the normality test at $\alpha = 0.05$, $p \leq 0.0001$). One

sample t-test revealed that the molecular weight between cyanobacterial proteomes was significantly different ($t = 31.24$, degree of freedom $df = 229$, $p = 0.0001$, $\alpha = 0.05$ (significant)).

## Cyanobacterial proteins ranged in size from 1.4007 to 1200.393 kDa

The molecular weight of individual cyanobacterial proteins ranged from 1.4007 to 1200.393 kDa (S1 File). *Oscillatoria nigro-viridis* was found to encode the smallest cyanobacterial protein, MAVISVTAATNLIP (1.40 kDa, accession WP_071884041.1), while *Anabaena cylindrica* was found to encode the largest cyanobacterial protein with a molecular weight of 1200.393 kDa (accession WP_015217688.1). A tandem-95 repeat protein was found in the largest protein in the cyanobacterial proteome. The average molecular weight of cyanobacterial proteins was 34.647 kDa. At least 80 proteins from among the 903149 analyzed protein sequences were found to have a molecular weight > 500 kDa (0.008%), and at least 29671 (3.28%) protein sequences were found to have a predicted molecular weight of 100–500 kDa (S1 File). Approximately 16.298% of the proteins had a predicted molecular weight in the range of 50–100 kDa, while 69.15% of the protein had a molecular weight of 10–50 kDa, and 11.26% had a molecular weight of 1–10 kDa (S1 File). Some of the high molecular weight cyanobacterial proteins included a PKD domain-containing protein (1139.1315 kDa, accession WP_096661492.1) and a non-ribosomal peptide synthetase (914.085 kDa, accession WP_100898072.1), RHS family protein (833.607 kDa, accession WP_096661480.1), as well as several others. At least 101712 (11.26%) protein sequences encoded a protein with a predicted molecular weight of ≤10 kDa. A few of the low-molecular-weight cyanobacterial proteins were, MGLLCGIWLRRKN (1.559 kDa, accession PZV24433.1), MGVSSLASRLVNCNI (1.563 kDa, accession CEJ46831.1), MTGHSLTIDGGYTVQ (1.579 kDa, accession WP_094672089.1), MSITEIIDDFPELT (1.623 kDa, accession WP_094673095.1), and MRNPVGSTHITASKDG (1.670 kDa, accession WP_081980801.1).

## The range of Cyanobacterial proteins is 180.617 to 480.131 amino acids per protein

The number of amino acids in the identified cyanobacterial proteins ranged from 180.617 to 480.131 amino acids per protein sequence. Longer proteins possessed a greater number of conserved functional domains and a greater number of predicted biological functions, while shorter proteins possessed fewer conserved domains and predicted biological functions. The greatest average length of protein sequences was found in *Microcoleus* sp. PCC7113 (480.131), while the lowest average length was found in *Trichodesmium erythraeum* (181.617). Collectively, the average number of amino acids present per cyanobacterial protein was 312.018.

## Cyanobacterial proteome encodes a greater number of acidic *pI* proteins

The *pI* of cyanobacterial proteins ranged from 2.13 to 13.32. The BEN50 protein (accession number PNW56779.1) from *Halothece* sp. (accession number WP_036263155.1) was found to have the lowest predicted *pI* of 2.13, while a protein sequence (accession number WP_036263155.1) from *Mastigocoleus testarum* had the highest *pI* (13.32) (S1 File). The overall average *pI* of cyanobacterial proteins was 6.437 (median 6.419) (Fig 1). Approximately 33.57% of protein sequences from among the 903149 analyzed protein sequences were found to have a *pI* in the basic range. In comparison, 66.30% of protein sequences had a *pI* in the acidic range. Only 0.12% of protein sequences were found to have a neutral *pI* (S1 File). The overall average of basic *pI* proteins was 8.629 (Fig 1), while the overall average of acidic *pI* proteins was 5.296 (Fig 1). Interestingly, a few of the high *pI* proteins were found to contain repetitive amino acids. The highest *pI* encoding protein (accession: WP_036263155.1, *Mastigocoleus*

*testarum*) possessed six R-R-R repeats with 50% of Arg amino acids. Similarly, a protein (accession: WP_084739217.1 from *Chroococcidiopsis thermalis*) with a *pI* of 13.1 was found to encode six G-T-R-G repeat sequences. The 50S ribosomal protein L34 (*Cyanobium* sp. ARS6) with a *pI* 13.01 contained R-R-R, R-R-V, and R-R-K repeats. In contrast, a protein (accession number WP_052324745.1) from *Hassallia byssoidea* was found to encode eleven M-K-L-R-V repeat sequences. We analyzed the amino acid composition of all the protein sequences with a *pI* ≥ 13 and found that they do not possess Asp, Cys, Glu, His, Phe, Trp, or Tyr. A protein (accession number WP_047157505.1) in *Trichodesmium erythraeum* had a predicted *pI* of 2.181 and contained at least 15% Asp and 6.4% Glu amino acids, which are both negatively charged. The amino acid composition of protein sequences with a *pI* ≤ 2.30 did not contain Cys, His, or Lys. To better understand the group-specific *pI* distribution of cyanobacteria, we grouped the cyanobacterial species into different groups and analysed the *pI*. The increasing order of *pI* of cyanobacterial species belonged to different order Spirulinales (6.135), Oscilla-toriales (6.263), Chroococcales (6.342), Synechococcales (6.344), Pleurocapsales (6.397), Nos-tocales (6.494), Unclassified (6.553), Chroococcidiopsidales (6.578), Gloeobacterales (6.734), and Melainabacteria (7.073) (S1 File).

A correlation analysis was conducted with GC% content and *pI* of cyanobacterial proteome. It was found that GC% and pI of cyanobacterial proteome is slightly correlated ($r = 0.2171$) (Fig 2). A comparative evolutionary study of the *pI* of cyanobacterial proteome revealed, *Roseofilum reptotaenium* proteome exhibited the lowest *pI* i.e., 5.893 evolved approximately 2180 million years ago (S1 Table). Similarly, the proteome of *Candidatus gaganbacteria* exhibited the highest *pI* i.e., 7.388 was evolved 1426–2635 million years ago. All the cyanobacterial proteomes that possess *pI* of more than 7.10 evolved 1426–2635 million years ago (S1 Table). Whereas few of the cyanobacterial proteome that contains *pI* 5.9–5.97 were found to evolve approximately 452–1157 million years ago (S1 Table).

## The Molecular weight and *pI* of Cyanobacterial proteins exhibit a bimodal distribution

The molecular weight and isoelectric point of cyanobacterial proteins greatly vary among the different proteomes. Notably, a bimodal distribution of cyanobacterial proteomes is evident (Fig 3) that deciphers the virtual 2D map of cyanobacterial proteomes. The overall average *pI* of cyanobacterial proteomes was 6.437, while the overall average molecular weight was 34.7417 kDa. The variance of the *pI* was found to be 0.068, and the variance of molecular weight was found to be 7.992. Variances lower than the mean reveal the binomial distribution of *pI* and molecular weight. Correlation analysis results indicated that cyanobacterial proteins' molecular weight and isoelectric point are negatively correlated ($r = -0.197$) (S2 Fig). The correlation analysis of amino acid sequence length of cyanobacterial proteins and *pI* also exhibited a negative correlation ($r = -0.240$) (S3 Fig).

The normal distribution analysis of *pI* for probability P(X > 13.32), P(X < 13.32), P (X > 2.13), and P (X < 2.13) were 0, 1, 0.992, and 0.0078, respectively. The probability of a *pI* with P(X > 7) was 0.424 and P(X < 7) was 0.575. This indicates that the probability of finding a protein with a *pI* > 13.32 in cyanobacteria is zero, while the probability of finding a protein with a *pI* < 2.13 is 0.0078. Similarly, the normal distribution analysis of molecular weight for probability P(X > 1200.393), P(X < 1200.393), P(X > 1.400), and P(X < 1.400) were 0, 1, 0.872, and 0.127, respectively. This indicates that the probability of finding a cyanobacterial protein with a molecular weight > 1200.393 kDa is zero, while the probability of finding cyanobacterial protein with a molecular weight > 1.400 is 0.872, and the probability of finding a protein with a molecular weight < 1.400 is 0.127. At least 177 species of cyanobacteria were

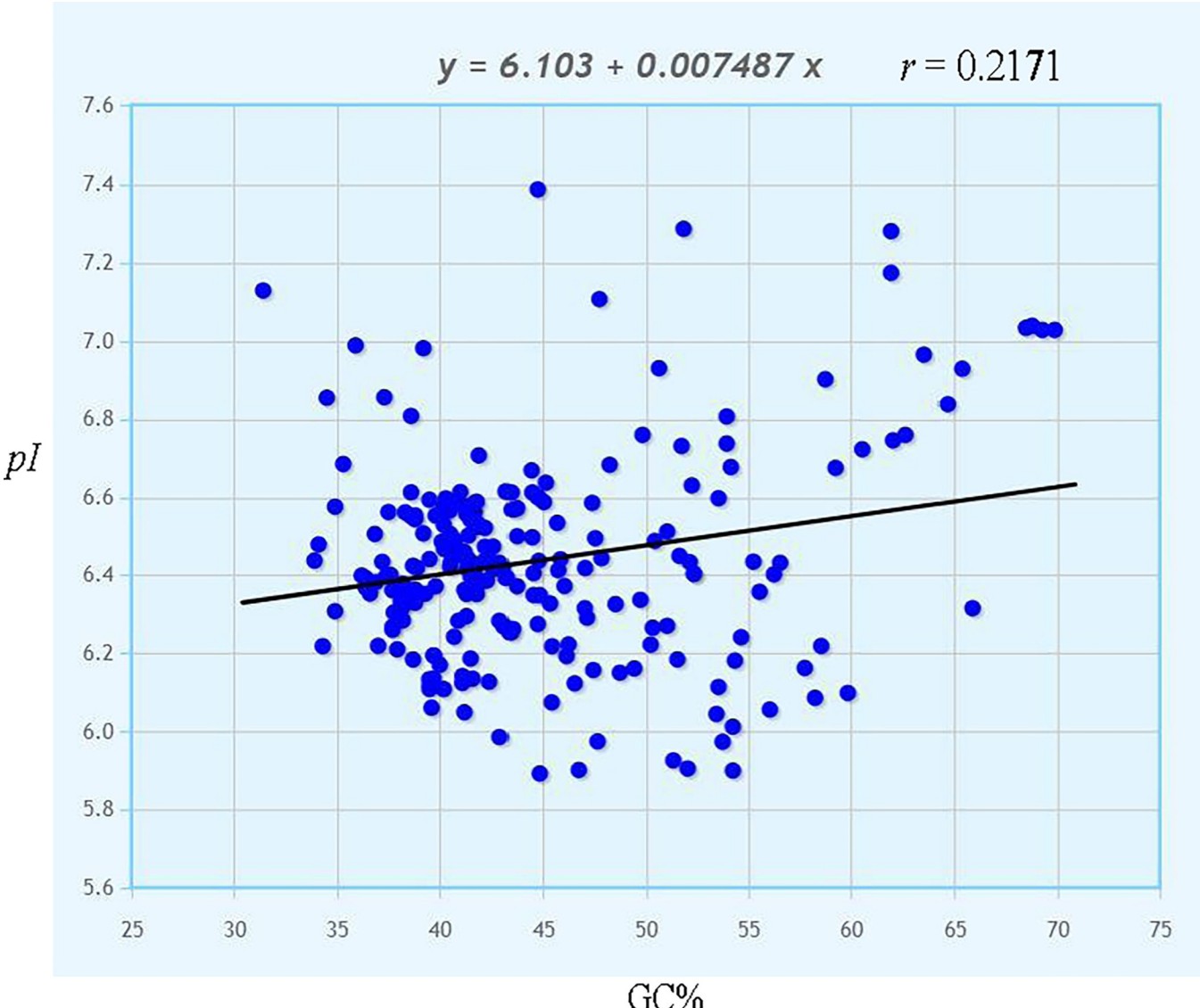

**Fig 2. Correlation analysis of GC% with *pI* of cyanobacterial proteome.** GC% show positive correlation *r* = 0.2172 with *pI*. However, the correlation coefficient was not so significant. The photographs was generated using mathportal server https://www.mathportal.org/calculators/statistics-calculator/correlation-and-regression-calculator.php.

found to encode neutral *pI* proteins. Gamma distribution of isoelectric point of cyanobacterial proteome showed empirical probability closely matches with the theoretical probability at 95% confidence interval (S4 Fig).

## Highest and lowest represented amino acids in the Cyanobacterial proteome

Proteome-wide analysis of the cyanobacterial proteome revealed that Leu (11.104%) was the most abundant and Cys (1.014%) was the least abundant amino acid (Table 1). Other highly abundant amino acids included Ala (8.324%), Gly (6.837%), Ile (6.619), and Val (6.536%). Low abundant amino acids, in addition to the Cys, included Trp (1.436%), His (1.87%), Met (1.88%), and Tyr (2.983%) (Table 1, Fig 4). Approximately 51.462% of the cyanobacterial

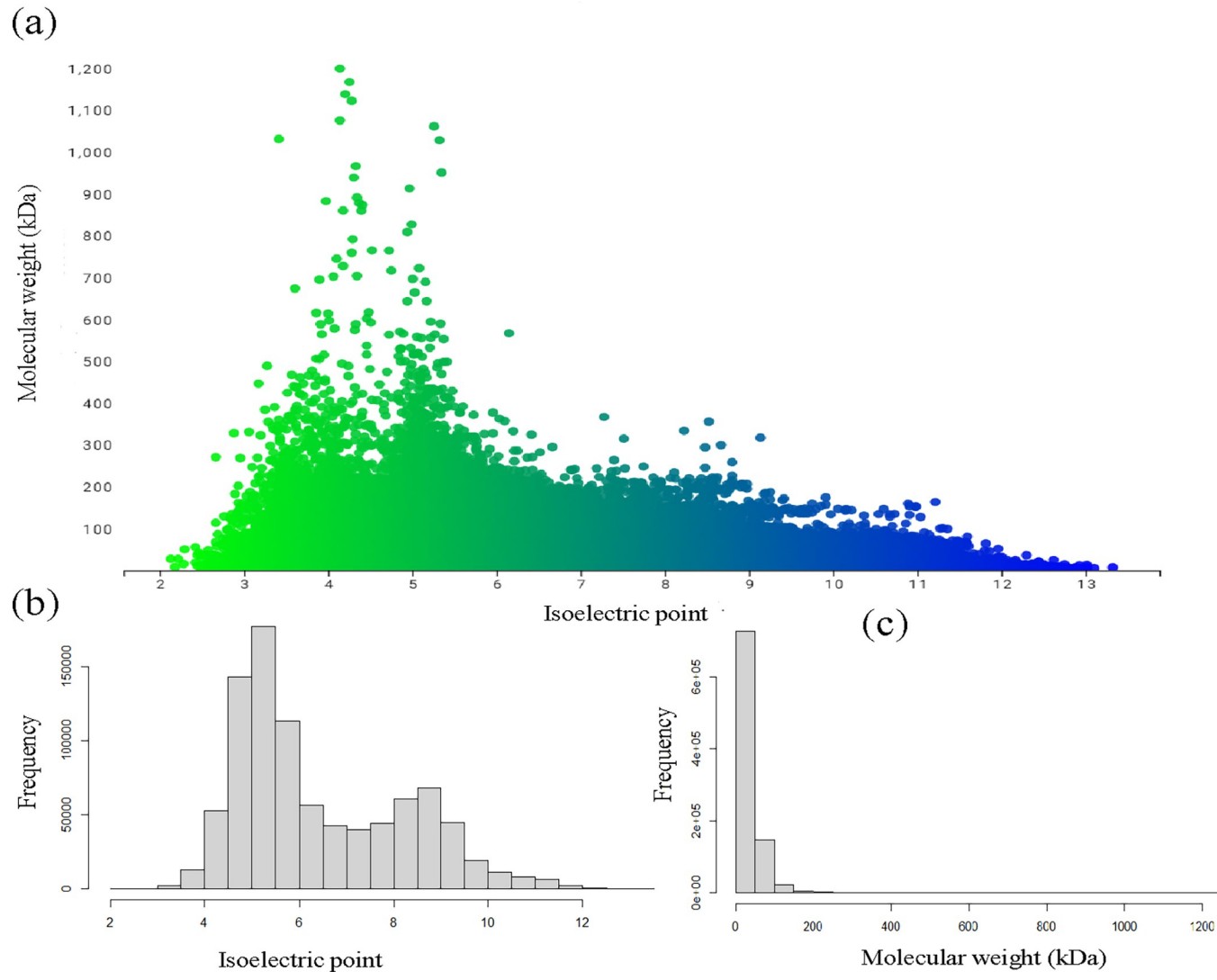

**Fig 3.** (a) Virtual 2D map of cyanobacterial proteome. X- axis represents isoelectric protein and Y-axis represents molecular weight (kDa). (b) Represents the frequency of isoelectric point of proteins and (c) represents the frequency of molecular weight of cyanobacterial proteins. The scatter plot was generated using scatterplot online software https://scatterplot.online/.

proteome contained nonpolar amino acids and 48.525% polar amino acids. The highest and lowest abundant amino acids in different cyanobacterial species were also calculated. *Prochlorococcus marinus* contained the highest percentage of Asn (6.467%), Phe (4.966%), and Ser (7.722%) amino acids, while *Candidatus gastranaerophilales* contained the lowest percentage of Arg (3.349%), Gly (5.926%), Leu (8.777%), and Pro (3.274%) amino acids (Table 1). *Aphanocapsa feldmannii* possessed the highest percentage of Cys (1.382%) and Arg (8.220%) amino acids, while *Aphanothece minutissima* contained the highest percentage of Gly (9.232%) and Pro (6.490%) amino acids. *Gloeomargarita lithophora* contained the highest percentage of Gln (6.325%) and Trp (1.845%) amino acids, and *Candidatus gastranaerophilales* encoded the highest percentage of Tyr (3.963%) and Lys (9.122%) amino acids (Table 1). The lowest percentage of Arg (3.349%), Gly (5.926%), Leu (8.777%), and Pro (3.274%) was found in *Candidatus gastranaerophilales*, while the lowest percentage of His (1.501%) and Ala (5.418%) was

**Table 1. Average amino acid composition of cyanobacterial proteomes.** Cyanobacterial species with highest and lowest abundance of amino acids are depicted here.

| Amino acids | Average percentage (%) | Highest percentage (%) | Name of the species with Highest percentage (%) | Lowest percentage (%) | Name of the Species with Lowest Percentage (%) |
|---|---|---|---|---|---|
| Ala | 8.324 | 12.471 | *Vulcanococcus limneticus* | 5.418 | *Prochlorococcus marinus* |
| Arg | 5.328 | 8.220 | *Aphanocapsa feldmannii* | 3.349 | *Candidatus gastranaerophilales* |
| Asn | 4.219 | 6.467 | *Prochlorococcus marinus* | 1.995 | *Cyanobium gracile* |
| Asp | 5.005 | 6.209 | *Leptolyngbya valderiana* | 4.286 | *Synechococcus sp.* 65AY640 |
| Cys | 1.014 | 1.382 | *Aphanocapsa feldmannii* | 0.890 | *Euhalothece sp.* KZN001 |
| Gln | 5.265 | 6.325 | *Gloeomargarita lithophora* | 2.910 | *Candidatedivision* WOR-1 |
| Glu | 6.263 | 7.434 | *Euhalothece sp.* KZN001 | 5.250 | *Candidatus synechococcus* |
| Gly | 6.837 | 9.232 | *Aphanothece minutissima* | 5.926 | *Candidatus gastranaerophilales* |
| His | 1.870 | 2.635 | *Candidatus synechococcus* | 1.501 | *Prochlorococcus marinus* |
| Ile | 6.619 | 9.232 | *Candidatus margulisbacteria* | 3.725 | *Vulcanococcus limneticus* |
| Leu | 11.104 | 13.225 | *Vulcanococcus limneticus* | 8.777 | *Candidatus gastranaerophilales* |
| Lys | 4.665 | 9.122 | *Candidatus gastranaerophilales* | 1.800 | *Synechococcus sp.* BO8801 |
| Met | 1.880 | 4.494 | *Tolypothrix bouteillei* | 1.508 | *Gloeobacter kilaueensis* |
| Phe | 3.895 | 4.966 | *Prochlorococcus marinus* | 2.923 | *Vulcanococcus limneticus* |
| Pro | 4.831 | 6.490 | *Aphanothece minutissima* | 3.274 | *Candidatus gastranaerophilales* |
| Ser | 6.330 | 7.722 | *Prochlorococcus marinus* | 4.893 | *Gloeomargarita lithophora* |
| Thr | 5.583 | 6.899 | *Tolypothrix bouteillei* | 4.219 | *Synechococcales bacterium* |
| Trp | 1.436 | 1.845 | *Gloeomargarita lithophora* | 0.005 | *Tolypothrix bouteillei* |
| Tyr | 2.983 | 3.963 | *Candidatus gastranaerophilales* | 1.759 | *Synechococcus sp.* BO8801 |
| Val | 6.536 | 7.543 | *Gloeobacter violaceus* | 1.445 | *Tolypothrix bouteillei* |

found in *Prochlorococcus marinus*. *Tolypothrix bouteillei*, however, had the lowest percentage of Trp (0.005%) and Val (1.445%), while *Vulcanococcus limneticus* had the lowest percentage of Ile (3.725%) and Phe (2.923%) (Table 1). A comparative study of the highest and lowest abundant amino acid-containing species revealed, the Cyanobacterial species *Vulcanococcus limneticus*, *Prochlorococcus marinus*, *Euhalothece sp. KZN001*, *Gloeomargarita lithophora*, *Candidatus gastranaerophilales*, and *Candidatus synechococcus* have both the highest and lowest abundant amino acids in their proteome (Table 1). A principal component analysis (PCA) revealed that Arg, Pro, Asp, Gln, Thr, Glu, Ser, Val, and Gly clustered together. In contrast, Trp, His, Met, and Cys clustered in a separate group (Fig 5). The high-abundant amino acids, Leu, Ala, and Ile, were located separately and independent from the other two clusters in the PCA plot. Trp, His, Met, and Cys are comparatively low-abundant amino acids in the cyanobacterial proteome and were found to cluster together in the PCA plot (Fig 5). A correlation plot of amino acid composition revealed a strong correlation ship between some amino acids (Fig 6A). Phe-Tyr (0.996), Leu-Thr (0.992), Gly-Pro (0.993), Leu-Gly (0.994), and Asn-Lys (0.992) were among the strongly correlated amino acid pairs. Amino acid pairs with poor correlations included Met-Trp (0.849), Ala-Lys (0.836), Ala-Asn (0.862), Arg-Ile (0.892), Arg-Lys (0.839), and Lys-Trp (0.876) (Fig 6). Network plot also shows a strong correlation of Trp-Val (0.99), and Arg-Ala (0.989) (Fig 6B).

Notably, the proteome-wide analysis of amino acid composition also revealed the presence of selenocysteine (Sec in the cyanobacterial proteome. *Candidatus melainabacteria* (accession PWT95605.1) was found to encode one Sec amino acid in its proteome. No other species, however, contained proteomes that encoded Sec amino acids. Notably, several proteins were annotated with the term "seleno", such as tRNA 2-selenouridine synthase, selenocysteine lyase, selenophosphate synthase, and selenocysteine-specific translation elongation factor. None of these proteins, however, were found to encode any Sec amino acids. A correlation study of

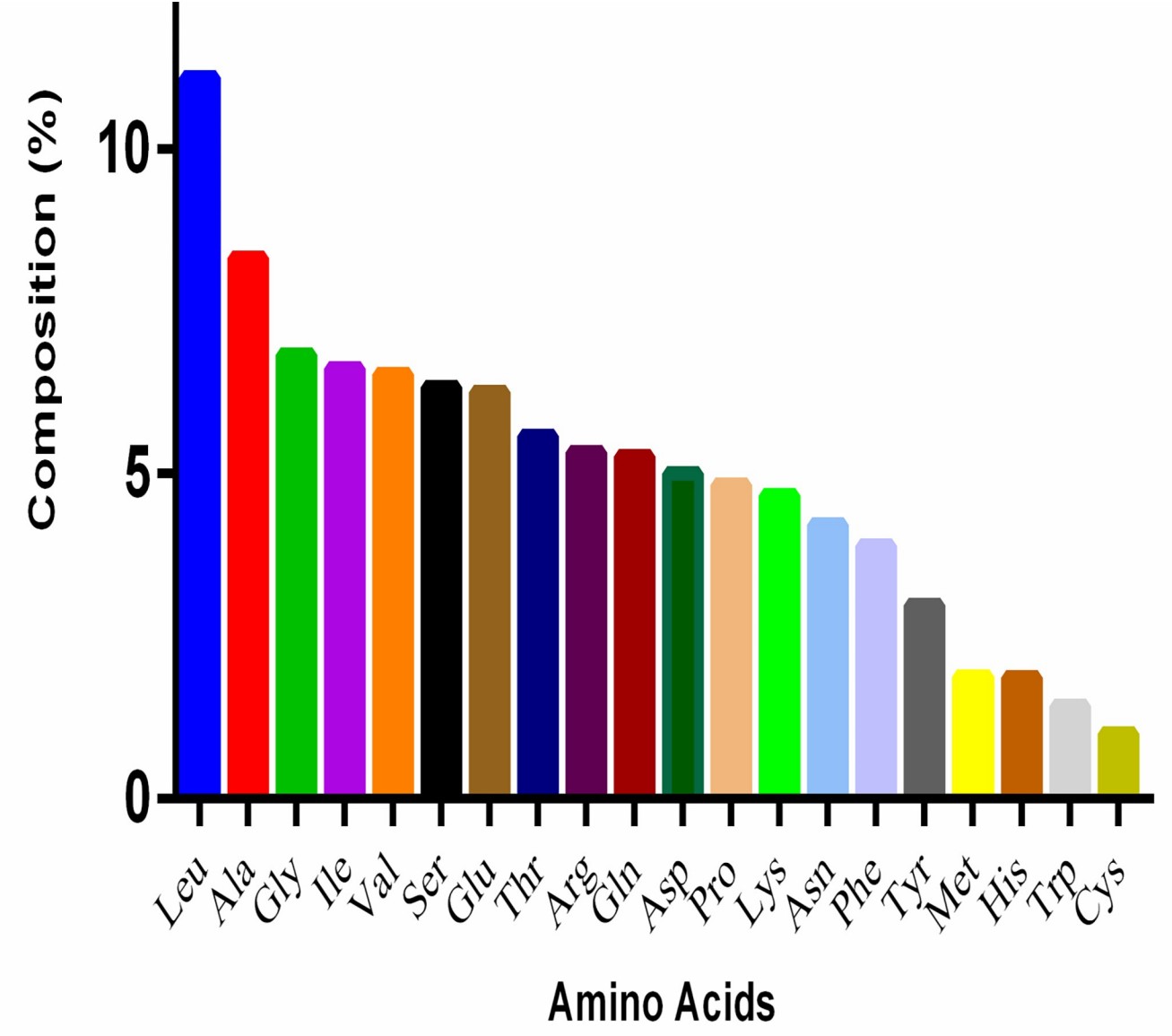

**Fig 4. Average amino acid composition of cyanobacterial proteomes (%).** Figure shows Leu is the highest and Cys is the lowest abundant amino acid in the cyanobacterial proteome.

Cyanobacterial GC% content was conducted with the amino acid composition. The study revealed that the GC% content of the cyanobacterial genome with amino acid usage was not so significant (Table 2). The highest correlation was found in the case of Met (0.0966), whereas the lowest correlation was found in the case of Trp (-0.008) (Table 2). Met (CAT) and Trp (CCA) is encoded by a single isoacceptor, and this might be the reason why CAT and CAU have the highest and lowest correlation coefficient with GC%. However, other amino acids are encoded by more than one isoacceptor [34–37].

## Discussion

The completion of the sequencing of several cyanobacterial genomes has provided an excellent opportunity to analyse and better understand their genomic and proteomic details [38–40].

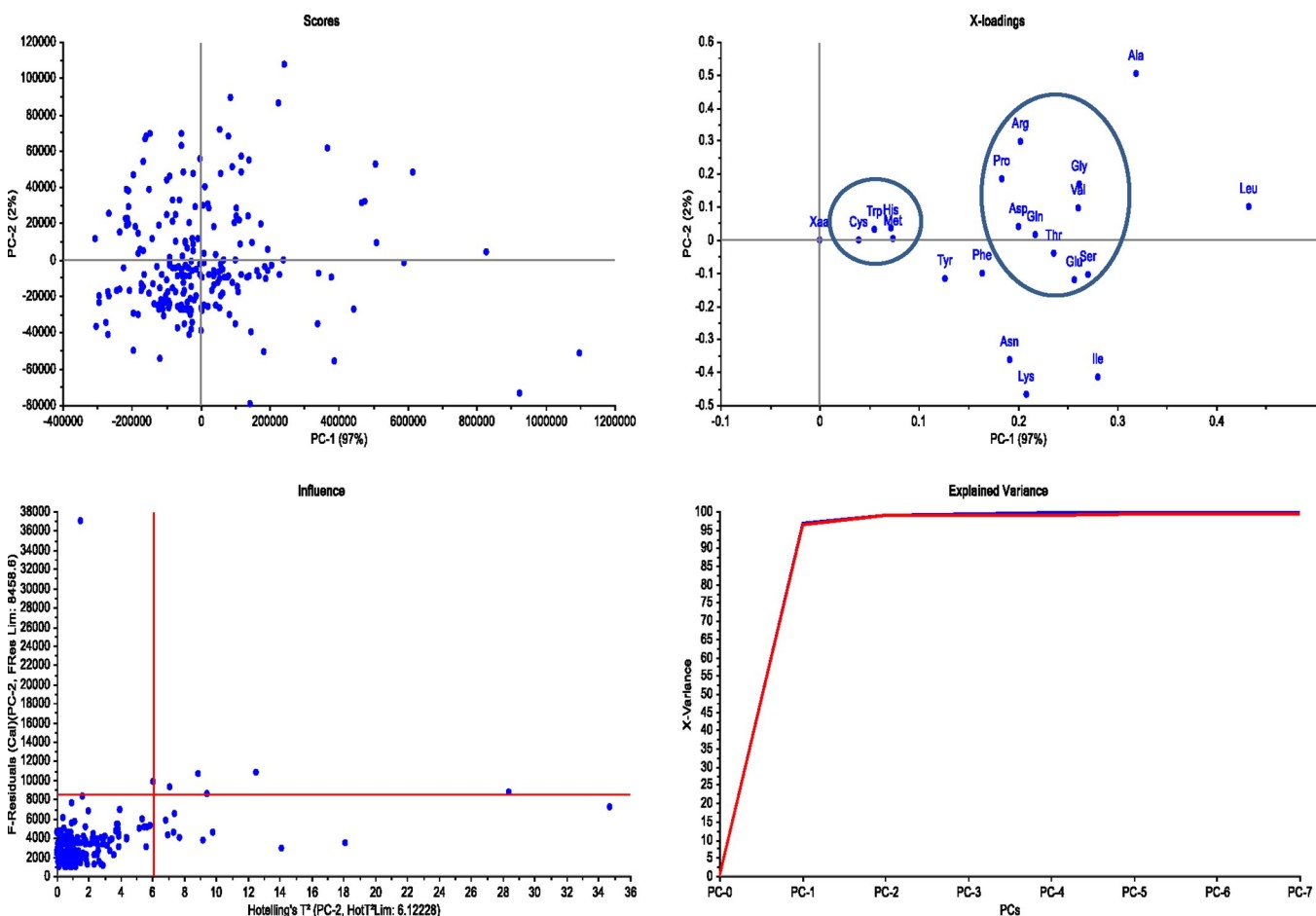

**Fig 5. Principal component analysis (PCA) of amino acid composition of cyanobacterial proteome.** The photograph was generated using Unsramber software version 7.

Although genome sequencing has provided important information on the genes and genome composition of the cyanobacteria, especially with regard to potential biotechnological applications [38], very little information is available with regards to its proteomic. Although cyanobacterial kingdom has not received enormous attention with regard to its proteomic study, it has still made a profound impact on the biotechnological implication of producing single-cell protein [41]. Therefore, it has become an excellent platform for understanding the proteomic details of the cyanobacteria and extracting information on the global identification of expressed proteins in cyanobacterial cells, as well as providing valuable insights into the dynamic response of cyanobacterial cell's environmental challenges and the regulation, compartmentalization, structure, and biological function of expressed proteins [42,43]. However, it is a challenging goal to assess and characterize the complete proteome of the entire cyanobacterial kingdom, and researchers have worked diligently to achieve this goal. The major core proteome of the cyanobacteria kingdom constitutes a few major protein families that vary dynamically among and between the species [38]. In this regard, we conducted a proteome-wide analysis in the present study by downloading and analysing the annotated protein sequences of all of the available cyanobacterial proteins, covering 229 cyanobacterial species (S2 File). In its entirety, our study collected and analyzed 903149 cyanobacterial protein sequences and constructed a virtual 2D map of the cyanobacterial proteome based on the

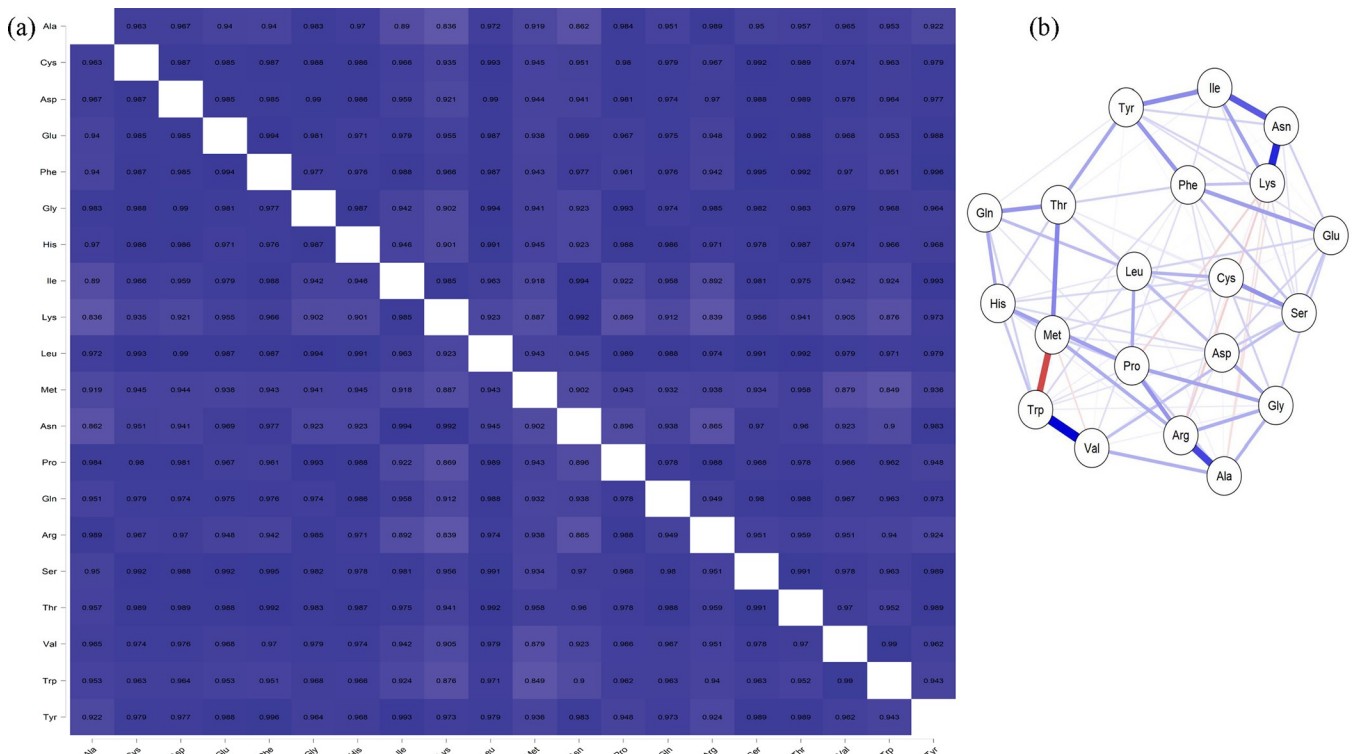

**Fig 6.** (a) Correlation plot of amino acid compositions and (b) network plot of amino acid composition of cyanobacterial protein. Strong and thick blue line indicates a positive correlation. The photographs was generated using mathportal server https://www.mathportal.org/calculators/statistics-calculator/correlation-and-regression-calculator.php.

molecular weight and isoelectric point of each of the collected protein sequences (S1 File). Among other results [44–47], the current analysis revealed the bimodal distribution of the molecular weight and isoelectric point of cyanobacterial proteins. The presence of higher percentage of polar amino acid at the surface of the protein and non-polar amino acids at the core of the protein result in increased isoelectric point conferring a greater thermostability [48,49]. Lower isoelectric point and minimum negative energy play important role in stabilization of bonds in acidic environment [50,51]. We also documented those cyanobacterial proteins contain an average of 312.018 amino acids per protein sequence. The molecular weight of proteomes from the order Spirulinales (37.151 kDa) was found to be the highest, whereas the molecular weight of proteomes from unclassified (30.519 kDa) cyanobacterial group was found to be the lowest. However, Chroococcidiopsidales (34.311 kDa), Chroococcales (34.487 kDa), Gloeobacterales (34.72 kDa) were found to encode proteomes within the range of 34 kDa. The phenotypic shape of the cyanobacteria from the order Chroococcidiopsidales and Chroococcales are coccoid, and the similar cellular structure might be the cause of encoding the proteome with approximately similar molecular weight.

A previous study reported that higher plant proteins contain an average of 423.34 amino acids per protein [44], which is significantly higher than the average found in cyanobacterial proteins. It was also previously reported that plants encode 40469.83 protein sequences per species, while fungi encode 10345.83 protein sequences per species [44,46]. Cyanobacteria, however, were found to encode only 5260.704 protein sequences per species. Although the virtual 2D map of the cyanobacterial and fungal [46] proteome exhibit a bimodal distribution for *pI* and molecular weight, the virtual 2D map of the plant proteome exhibits a trimodal

**Table 2. Correlation analysis of GC% content and amino acid usage bias in cyanobacterial proteome.**

| Amino acids | Regression equation Y | Correlation coefficient *r* |
|---|---|---|
| Ala | 114917 + 442 x | 0.0518 |
| Cys | 14174 + 49.43 x | 0.0484 |
| Asp | 74508 + 160.9 x | 0.0308 |
| Glu | 94837 + 173.2 x | 0.0259 |
| Phe | 58337 + 126.4 x | 0.0296 |
| Gly | 96186 + 318.4 x | 0.0466 |
| His | 26600 + 83.05 x | 0.044 |
| Ile | 104749 + 91.86 x | 0.0123 |
| Lys | 72252 + 103.6 x | 0.0181 |
| Leu | 164555 + 347.1 x | 0.0309 |
| Met | 21818 + 195.8 x | 0.0966 |
| Asn | 66215 + 86.4 x | 0.0166 |
| Pro | 67356 + 245.3 x | 0.0507 |
| Gln | 80610 + 140.5 x | 0.0246 |
| Arg | 72412 + 304.9 x | 0.0563 |
| Ser | 97274 + 153.9 x | 0.0219 |
| Thr | 83054 + 202 x | 0.0329 |
| Val | 102743 + 82.97 x | 0.0121 |
| Trp | 23895–13.01 x | -0.008 |
| Tyr | 44761 + 98.37 x | 0.0297 |

distribution [44]. The virtual 2D map of virus proteomes showed host-specific modalities, molecular weight, and isoelectric points [47]. Like the cyanobacteria, the *pI* of most proteins encoded in the plant and fungal proteome reside in the acidic *pI* range [44,46]. Although cyanobacteria are found in diverse habitats [52,53], the *pI* of their proteins primarily resides in the acidic range. A few species of Cyanobacteria, however, were found to exhibit an average *pI* of their overall proteome in the basic range. These species include *Candidatus gaganbacteria* (7.388), *Aphanothece minutissima* (7.028), *Candidate division* WOR-1 (7.287), *Candidatus synechococcus spongiarum* (7.174), *Candidatus termititenax* (7.107), *Cyanobacterium* PCC 7702 (7.033), *Cyanobium gracile* (7.039), *Prochlorococcus marinus* (7.129), *Synechococcus* sp. BO8801 (7.028), and *Vulcanococcus limneticus* (7.033) (S1 Table). The species *Vulcanococcus limneticus* was isolated from a volcanic lake in central Italy. It was found to contain the highest percentage of Ala (12.471) and Leu (13.225) amino acids among the studied cyanobacterial species, whereas it contained the lowest percentage of Phe (2.923) (Table 1). It might need the highest percentage of Ala and Leu amino acids to withstand higher temperature, and hence V. limneticus contained the highest percentage of Ala and Leu amino acids. Similarly, Phe might be the least required amino acid to withstand higher temperatures. *Prochlorococcus marinus* is a picoplankton that shows unusual pigmentation due to the presence of chlorophyll a2 (a derivative of chlorophyll a) and b2. It contained the highest percentage of Phe (4.966) and Ser (7.722) amino acids. *P. marinus* does not contain Chl a as a major photosynthetic pigment but contains α-carotene [54]. Similarly, P. marinus contained the lowest percentage of Ala (5.418) and His (1.501) amino acids. The composition of the highest and lowest abundance amino acids might play an important role in the lack of Chla and having Chl a2 for photosynthesis. A few of the cyanobacterial species exhibiting a proteome with a low average *pI* were *Roseofilum reptotaenium* (5.893), *Aphanocapsa montana* (5.974), *Euhalothece* sp. KZN001(5.902),

*Halomicronema excentricum* (5.900), *Halothece* sp. PCC7418 (5.986), *Oscillatoria acuminata* (5.975), *Phormidium lacuna* (5.926), and *Phormidium* sp. SL48-SHIP (5.905) (Table 1). These data indicate that cyanobacterial proteomes exhibit a wide range of average *pI*. When we compared the amino acid composition with the plant proteomes, we found that Cys (1.014%) was the lowest encoding amino acid in the Cyanobacterial kingdom whereas Trp (1.28%) was the lowest encoding amino acid in the plant kingdom [44]. However, the abundance of Leu amino acid was highest in the Cyanobacteria and plant kingdom.

Our analysis also revealed several highly repetitive proteins in some cyanobacterial proteomes, all of which had a *pI* < 3. The accession number of some of these repetitive proteins with a *pI* < 3 were WP_079680752.1, WP_045868631.1, WP_041565552.1, WP_041234656.1, WP_040484803.1, WP_039747676.1, WP_036265112.1, WP_035758608.1, WP_027842305.1, WP_017716411.1, WP_015226612.1, WP_015178523.1, WP_015156103.1, WP_006634710.1, and WP_006194633.1. Evolutionary analysis of cyanobacterial *pI* revealed the lowest *pI* encoding species, *Roseofilum reptotaenium* evolved 2180 million years ago. In contrast, the cyanobacterial proteome with *pI* > 7 was found to evolved 1426–2635 million years ago, suggesting the dominance of basic *pI* protein in the early stage of life. Similarly, the maximum cyanobacterial species containing *pI* 5.9 to 5.97 evolved 452–1157 million years ago. This suggests that the dominance of acidic *pI* proteome in the cyanobacterial lineage is a recent event (S1 Table). This confirms that the evolution of cyanobacterial proteome tends towards the acidic *pI*, and this might be associated with the event of ocean acidification [55–57]. Group-specific *pI* analysis revealed, Spirulinales (6.135) encoded the lowest *pI* proteins (acidic *pI* range), whereas Melainabacteria (7.073) encoded the highest *pI* proteins (basic *pI* range) (S1 File). The cyanobacterial groups that encoded *pI* in the range of 6.2 to 6.4 were Oscillatoriales (6.263), Chroococcales (6.342), Synechococcales (6.344), and Pleurocapsales (6.397). It was important to note that, Spirulinales (37.151 kDa) and Oscillatoriales (36.701 kDa) encoded high molecular weight proteins, and on the other hand, they contain low *pI* proteins. This suggests that increases in molecular weight of cyanobacteria is directly proportional to the decrease in the isoelectric point of the protein in the case of Spirulinales and Oscillatoria (S1 File). Melainabacteria is not able to perform photosynthesis [58], and hence they obtain energy by fermentation [59]. They lack an electron transport chain system and use Fe-Hydrogenase to produce hydrogen gas [58]. However, Melainabacteria has the potential to fix nitrogen [58].

A comparison of the molecular weight of the cyanobacterial and fungal proteome revealed that the average molecular weight of fungal proteins is 50.90 kDa, while the average molecular weight of cyanobacterial proteins is 34.647 kDa. Cyanobacteria are prokaryotic organisms and lack non-coding genomic DNA but still contain smaller-sized proteins. The biological function of a protein is partially determined by its three-dimensional tertiary structure, which is directly affected by the primary structure of the polypeptide chain of amino acids [60,61]. Longer peptides have greater possibilities for accommodating multiple structural and functional domains. A comparative study of protein size from a limited number of taxa revealed considerable differences in protein size [61]. More than 90% of *E.coli* K-12 strain lies in the small isoelectric point and molecular weight of 4–100 kDa, which is in accordance with the cyanobacterial proteome [62]. Some of the 2-DE isoform spots of the same gene had different *pI*, suggesting the role of post-translational modification of the protein [62]. However, most of the predicted and observed molecular weight and isoelectric point of *E. coli* K-12 strain showed reasonable correlation [62], suggesting the significance of our study.

Eukaryotic proteins were reported to possess proteins that, on average, are longer than bacterial proteins, which in turn have a longer average length than archaeal proteins [61,63]. Larger proteins in prokaryotic organisms have been reported to reflect the evolution toward larger protein size [61,63,64]. However, the average protein size in *Chlamydomonas*

*reinhardtii*, *Volvox carteri*, and other lower eukaryotic organisms is larger than the average protein size of higher plant species [44,65,66]. The evolution of eukaryotic proteins has been reported to have occurred via the fusion of single-function proteins into multi-domain and multi-functional proteins [63]. The fusion of domains has increased the average size of proteins. It can potentially lead to a reduction in the number of individual proteins in the proteome of a species and its corresponding number of individual genes. Although this is true for prokaryotic and eukaryotic lineages, it is not completely true for plant lineages, as unicellular photosynthetic organisms contain larger average-size proteins. Eudicot plants, however, are considered evolutionarily older than monocot plants, and the average protein size of monocot species is slightly larger than dicot species [44]. The average protein size in monocot plants has been reported to be 431.07 amino acids and 424.30 amino acids in dicot plants [44].

According to the starter-set hypothesis, proteins are assumed to have originated from a small set of starter sequences called functional domains, having a length of 4, 15, or 50 amino acids, which later expanded through gene duplication and other genetic modifications [67–70]. This hypothesis also states that gene or exon duplication or fusion has existed since the beginning of the evolution of larger protein sequences [64]. The random-origin hypothesis, however, states that proteins emerged from a large number of random heteropeptides [64,71–75]. The random-origin hypothesis also states that the existence of larger proteins has occurred by chance [74,75]. The presence of the largest protein (Tandem-95 repeat protein, accession WP_015217688.1) in the cyanobacterium, *Anabaena cylindrica*, with a molecular weight of 1200.393 kDa, seems to fit the random-origin hypothesis, as this protein is not found in all cyanobacterial species.

Gamma-type distribution is frequently used to explain protein length [64,74], which assumes that protein sequences may have been exponentially distributed in random lengths. That protein folding and stability are length-dependent and that the potential for an increased number of biochemical activities increases with protein length. The gamma distribution analysis of amino acid sequence length of cyanobacterial proteins closely matches the empirical vs. theoretical value (confidence interval 95%), suggesting a close fit of Gamma distribution for amino acid sequence length/protein size in cyanobacteria (S5 Fig). Correlation analysis of the number of protein sequences vs. the size of the proteome revealed a significant positive correlation ($r$ = 0.918). Strong selective forces have been reported to play an essential role in the increase in the number of proteins of a given size above the average frequency [64,76]. The stability of a protein is highly dependent upon its length, and thus longer proteins possess a selective advantage and have the potential to drive the evolution of a proteome [77,78]. The genetic composition can be theoretically used to assess average protein size and distribution. A stop codon can occur stochastically after a start codon, and it is highly possible that the presence of larger protein-coding sequences will be less frequent than smaller protein-coding sequences. Hence, only 3.28% of proteins are encoded with a molecular weight > 100 kDa. However, the role of *pI* in the frequency of the distribution of protein sizes is largely unknown. It seems improbable that the genome encodes proteins with varied sizes without consideration of charge and sub-cellular localization. In contrast, it is more plausible that proteins also undergo strong selective pressure based on their charge (*pI*) and size.

## Conclusion and future perspectives

Analysis of the Cyanobacterial proteomes from 229 Cyanobacterial species revealed a bimodal distribution of the molecular weight and isoelectric point of the Cyanobacterial proteome. The map deduced from the molecular weight, and isoelectric point of the Cyanobacteria reflects a virtual 2D map of the Cyanobacterial proteome. These theoretical proteome profiles of the

cyanobacteria can be experimentally validated to understand the post-translational modification of the proteins and their role in the change in the isoelectric point of the protein. Post-translational modification can change the molecular weight and isoelectric point of the protein. However, reversible post-translational modification can have a minimal difference in the molecular weight and isoelectric point. The study can be applied to understand the biochemical characteristics of any particular protein and its subsequent localization and function. The amino acid composition study can be very useful in understanding the codon usage bias in the Cyanobacterial genome in the future. The codon usage bias will enable us to understand the molecular evolution through codon optimization. The localization of proteins within the cyanobacterial cells is also poorly understood. It is imperative to produce the sub-cellular proteome map to understand their biochemical and physiological process.

## Materials and methods

### Sequence downloads and calculation of molecular weight and pI

From the National Center for Biotechnology Information (NCBI), we downloaded all the protein sequences from 229 cyanobacterial species (S1 Table, S1 and S2 Files) and analyzed the proteomic details of these organisms. All the protein sequences belonged to the annotated ORF (open reading frame)/CDS (coding DNA sequences) of the respective gene. The Cyanobacterial species were from 10 different orders, namely Chroococcales, Chroococcidiopsidales, Gloeobacterales, Melainabacteria, Nostocales, Oscillatoriales, Pleurocapsales, Spirulinales, Synechococcales, and Unclassified (S1 Table). All the species were considered for the analysis available till January 2021. The repetitive protein sequences of the individual species were identified using the proteome file of the respective species. The molecular weight and isoelectric point of the protein sequences were calculated using a Linux-based isoelectric point (*pI*) calculator [79]. The script for the calculation of molecular weight and the isoelectric point is mentioned as mentioned below. The script was ipc <fasta_file> <pK$_a$ set> <output_file> <plot_file>. The pKa sets of the ipc can be found in the software itself. The calculated *pI* and molecular weight of each cyanobacterial protein were then processed using Microsoft Excel 2016. The molecular weight and isoelectric point of the proteins were used to construct the virtual 2D map of the proteome of Cyanobacteria using scatterplot online (https://scatterplot.online/).

Our team used a Linux-based program to calculate the amino acid composition of the cyanobacterial proteomes as well as the amino acid count in each protein sequence. Later we merged all the proteome files that contain information about amino acids to determine the amino acid composition of Cyanobacteria and the overall proteome as a whole.

### Statistical analysis

Several methods were used to analyse the derived molecular weight, isoelectric point, amino acid composition, and amino acid sequence length of the cyanobacterial proteins. In correlation analyses, the number of protein sequences in a proteome was compared to the average molecular weight of proteins, the weight of proteomes was compared to protein identity, and the amino acid sequence length was compared to protein identity. Based on amino acid composition, correlation and network plots were constructed, and the Gamma distribution of amino acid sequence length was calculated using JASP 0.14.1.0 software. JASP is an inbuilt software for statistical analysis. The user needs to submit the sample data in CSV file format. Once the data are uploaded, the user can choose the statistical option to run the analysis. A principal component analysis of amino acid composition was performed by Unscrambler 11. Leverage correction was done to conduct PCA analysis. The statistical analysis was conducted

at a 95% significance level ($p < 0.05$). Normal distribution explains the probability of occurrence of protein with *pI* 2.13 and 13.32 and molecular weight of 1.4 kDa and 1200.393 kDa. Therefore, we conducted a normal probability distribution of molecular weight and isoelectric point. Normal distribution was conducted using math portal (https://www.mathportal.org/calculators.php) calculator.

### Evolutionary time scale

The evolutionary time scale of the cyanobacterial species was estimated using the time tree: the time scale of life server (http://www.timetree.org/) [80]. In the time tree server, it needs to specify the taxon name to get the evolutionary timeline and divergence of the species.

## Supporting information

**S1 Fig. Correlation regression analysis of number of protein sequences with molecular weight of proteomes (kDa).**
(PPTX)

**S2 Fig. Molecular weight of proteomes and *pI* of cyanobacterial protein.**
(PPTX)

**S3 Fig. Correlation plot of amino acid sequence length vs isoelectric point.**
(PPTX)

**S4 Fig. Gamma distribution of isoelectric point of cyanobacterial proteome.**
(PPTX)

**S5 Fig. Gamma distribution of amino acid sequence length in cyanobacterial proteome.**
(PPTX)

**S1 Table. Table depicting the list of the cyanobacterial species with the number of protein sequences used in this study.**
(DOCX)

**S1 File. Accession number, protein name, molecular weight and isoelectric point of the cyanobacterial proteins.**
(XLSX)

**S2 File. Details about the species name Genebank accession number, GC% content and *pI* of cyanobacterial species.**
(XLSX)

## Author Contributions

**Conceptualization:** Tapan Kumar Mohanta.

**Data curation:** Tapan Kumar Mohanta.

**Formal analysis:** Tapan Kumar Mohanta, Yugal Kishore Mohanta.

**Investigation:** Tapan Kumar Mohanta.

**Methodology:** Tapan Kumar Mohanta.

**Project administration:** Tapan Kumar Mohanta.

**Resources:** Tapan Kumar Mohanta.

**Software:** Tapan Kumar Mohanta.

**Supervision:** Tapan Kumar Mohanta.

**Validation:** Tapan Kumar Mohanta.

**Visualization:** Tapan Kumar Mohanta, Satya Kumar Avula.

**Writing – original draft:** Tapan Kumar Mohanta.

**Writing – review & editing:** Tapan Kumar Mohanta, Satya Kumar Avula, Amilia Nongbet, Ahmed Al-Harrasi.

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
