## [Decision Letter · Decision Letter 0]

19 Aug 2022

PONE-D-22-12929Decoding the Virtual 2D Map of Cyanobacterial ProteomesPLOS ONE

Dear Dr. Mohanta,

Thank you for submitting your manuscript to PLOS ONE. After careful consideration, we feel that it has merit but does not fully meet PLOS ONE’s publication criteria as it currently stands. Therefore, we invite you to submit a revised version of the manuscript that addresses the points raised during the review process.Please submit your revised manuscript by Oct 03 2022 11:59PM. If you will need more time than this to complete your revisions, please reply to this message or contact the journal office at plosone@plos.org. Please include the following items when submitting your revised manuscript:A rebuttal letter that responds to each point raised by the academic editor and reviewer(s). You should upload this letter as a separate file labeled 'Response to Reviewers'.A marked-up copy of your manuscript that highlights changes made to the original version. You should upload this as a separate file labeled 'Revised Manuscript with Track Changes'.An unmarked version of your revised paper without tracked changes. You should upload this as a separate file labeled 'Manuscript'.

We look forward to receiving your revised manuscript.

Kind regards,

Arabinda Ghosh

Academic Editor

PLOS ONE

Journal Requirements:

Reviewers' comments:

Reviewer's Responses to Questions

**Comments to the Author**

1. Is the manuscript technically sound, and do the data support the conclusions?

Reviewer #1: Yes

Reviewer #2: Yes

Reviewer #3: Yes

2. Has the statistical analysis been performed appropriately and rigorously? 

Reviewer #1: Yes

Reviewer #2: Yes

Reviewer #3: Yes

3. Have the authors made all data underlying the findings in their manuscript fully available?

Reviewer #1: Yes

Reviewer #2: Yes

Reviewer #3: Yes

4. Is the manuscript presented in an intelligible fashion and written in standard English?

Reviewer #1: No

Reviewer #2: Yes

Reviewer #3: Yes

5. Review Comments to the Author

Reviewer #1: The manuscript submitted by Mohanta et al. has described the virtual 2D map of the cyanobacterial proteomes. They analyzed the Cyanobacterial proteomes of 229 species and reported the basic proteomics. The study contains each fundamental detail of the cyanobacterial proteome. It is promising research and holds high value get to publish in PLoS One. However, the article needs appropriate revision as per the comments provided below before acceptance for publication.

Overall comments; but the explanation should be reflected in the manuscript file to convey the importance of the study to the audience

1. Why authors have included 229 species in their study? Do they have some commonalities or chosen as random?

2. How the results of the cyanobacterial proteome can be corroborated with the bacterial or plant proteome. Please explain the similarities and differences, if any?

3. Is there any correlation with regard to molecular weight and isoelectric point of cyanobacterial proteomes?

4. Why Cys is lowest and Leu is highest abundant amino acid in the cyanobacterial proteomes. Although, it is difficult to establish the reason, do authors have any information or hypothesis behind this?

5. What is the possible reason for the presence of higher average basic pI proteins (8.62) in cyanobacteria?

Abstract:

I. Rewrite the sentence “…..The 20 genome sequence data of the Cyanobacteria has been extensively studied, but a detailed 21 proteomics study of the Cyanobacteria is lacking….” for better meaning and simplicity.

II. Rewrite the sentence “…..The Cyanobacterial proteome encoded a 27 greater number of acidic-pI proteins, and the average pI of the Cyanobacterial proteome was found 28 6.437…..”.

III. Write either full name or its abbreviation any one “molecular weight and pI”

IV. Pls. rewrite “…Using this computational analysis…..”

Introduction:

i. Make a single sentence “Therefore, proteomics provides more information about an organism compared to its genome. Proteomics provides an understanding of the cellular function and serves as a link between gene expression and translational products.”

ii. Rewrite the sentence “However, we lose lot of information about the whole proteome study as it is not possible to conduct 2D gel electrophoresis below pH 3 and above pH 11 due to lack of IPG (immobilized pH gradient) stripe below pH 3 and above pH 11.

iii. ………proteomic details can be written as ……..proteomics data…….

iv. Please rewrite this as both the consecutive sentences are started with Therefore….. line 81-86

The introduction section is nicely presented but moderate English editing will be more helpful to the diverse audience to understand this informative work.

Results

I. The results part is well described as per the aim and objectives of the Manuscript by the authors; however, the pictorial presentation would be more updated by increasing the resolutions.

II. The author should also mention the image source and the software used to make the images.

III. Units of the parameters should cross-checked.

Discussions

I. The authors have added sufficient discussion on their works and if within this time any more literature has been published, please add them in the discussion part which could give a complete frame for this work.

II. The introduction section is nicely presented but moderate English editing will be more helpful to the diverse audience to understand this informative work.

Conclusion

The author should add a paragraph on future prospects of these types of study in the application to the real scientific world. As this work is indeed a time taking analysis based on the data available in the database but it needs a very clear futuristic direction that can encourage following researchers

Materials and Methods

I. In this section, the Author should add a table providing details on databases (online/offline) and other internet sources which will be essential to carry out such type of study. It would be helpful for young budding researchers to handle such big data sets.

II. Authors should add abbreviations.

Moderate English language revision is a must prior to considering this manuscript for the publications.

Reviewer #2: - This study was designed similar to authors previous publication on ‘Virtual 2-D map of the fungal proteome’. This study used the sequences of 229 cyanobacterial species to analyse the molecular mass, average number of amino acids and isoelectric point of cyanobacterial proteome.

- This study focussed only on numbers and statistical data, and it lacks functional analysis.

- Following suggestions may improve the quality of manuscript.

- The whole structure of the manuscript needs to be reorganised.

- The headings in the results section seems like long sentences rather than section headings. It should be descriptive and as concise as possible.

- Table.1.: it can be provided as supplementary information. The information in this table can be segregated to identify the species with ascending or descending order of no of sequences/mol wt/pI. The functional analysis should be discussed in the discussion section. For example, the authors should extract the information on species with industrial applications and its correlation to above parameters.

- Table.2. Significance of highest and lowest percentage of amino acids and its correlation to different species should be discussed in the text.

- The manuscript should also focus the discussion on evolutionary time scale. Evolution of species and order may be represented in cladograms. Divergence and functional variations should also be discussed in the manuscript.

- Majority of the conclusion section is almost hypothetical. This paper not concluded anything relevant to the functional and structural analysis.

- Figure 6. B was not explained anywhere in the text. Figure legend should contain the information on colour patterns.

- Figure 5 legend should be precise. Remove the analysis part in figure legend

Reviewer #3: Decoding the Virtual 2D Map of Cyanobacterial Proteomes

The virtual 2D map of the cyanobacteria proteomes is reported in the manuscript. The authors reported the fundamental proteomics after analyzing the 229 species of cyanobacteria proteomes. Every essential aspect of the cyanobacteria proteome is covered in the study. The research is exciting and valuable enough to be published in PLoS One. Before being accepted for publication, the piece must, however, be appropriately revised in accordance with the remarks offered below.

Comments:

1. The title of the manuscript should be modified.

2. If possible graphical abstract should be provided.

3. Line no 412-414: Protein sequences of 229 cyanobacteria species were downloaded from NCBI and proteomics details of these organisms were analyzed.

4. The author should give a strong hypothesis before this study by taking huge data sets.

5. Why authors have considered 229 species and it will be better if the author would give a justification?

6. Why author has taken the parameter of analysis like molecular weight and isoelectric point in this study?

7. Line no 418 and 419: For analysis, recent data should be taken into account if available.

8. Line no 419 and 420: Only one proteome file was considered for 420 the analysis of those species contained repetitive protein sequence data. Is there any specific reason for that? If yes please mention it if possible.

9. Line no 420 and 421: The data analysis starting date was already mentioned, there is no need for repetition.

10. Line 427 and 428: Please provide the script if possible.

11. Line 437-439 and Line 446-447: Please explain how JASP Unscrambler 11, and time tree: the time scale of life server works.

12. In Fig 6 (b): Is there any particular explanation regarding different color coding between the interactions? If yes, please mention it.

13. All figures need a better resolution.

14. The author should also mention the image source and the software used to make the images.

15. The author must add a paragraph on future prospects

Recommendation: Major Revision

6. PLOS authors have the option to publish the peer review history of their article (what does this mean?). If published, this will include your full peer review and any attached files.

Reviewer #1: No

Reviewer #2: **Yes: **Vinod Kumar Yata

Reviewer #3: No

---

## [Author Response · Author response to Decision Letter 0]

23 Aug 2022

Dear editor,

Greetings

Please find the rebuttal letter to get point to point response.

Sincerely

Dr. Tapan

---

## [Decision Letter · Decision Letter 1]

8 Sep 2022

PONE-D-22-12929R1Virtual 2D Map of Cyanobacterial ProteomesPLOS ONE

Dear Dr. Mohanta,

Thank you for submitting your manuscript to PLOS ONE. After careful consideration, we feel that it has merit but does not fully meet PLOS ONE’s publication criteria as it currently stands. Therefore, we invite you to submit a revised version of the manuscript that addresses the points raised during the review process.

We look forward to receiving your revised manuscript.

Kind regards,

Arabinda Ghosh

Academic Editor

PLOS ONE

Journal Requirements:

Additional Editor Comments (if provided):

As per the reviewer suggestions there are few minor comments needs to be addressed before any decision on the manuscript.

Reviewer 1.

The authors have revised the manuscript very well and now acceptable for publication. However, I have a few minor comments that need to be looked after for future.

1. Do the correlation of pI and Molecular weight has any impact on structural aspects of the protein?

2. Can these Molecular weight and pI play role in structure of the protein?

Reviewer 2.

I agree with the authours response to my queries, and recommend this manuscript for publication

Reviewer 3.

The authors have addressed all the comments. However there are a very few grammatical errors should be addressed. I recommend this article for publication.

Reviewers' comments:

Reviewer's Responses to Questions

**Comments to the Author**

1. If the authors have adequately addressed your comments raised in a previous round of review and you feel that this manuscript is now acceptable for publication, you may indicate that here to bypass the “Comments to the Author” section, enter your conflict of interest statement in the “Confidential to Editor” section, and submit your "Accept" recommendation.

Reviewer #1: All comments have been addressed

Reviewer #2: All comments have been addressed

Reviewer #3: All comments have been addressed

2. Is the manuscript technically sound, and do the data support the conclusions?

Reviewer #1: Yes

Reviewer #2: Yes

Reviewer #3: Yes

3. Has the statistical analysis been performed appropriately and rigorously? 

Reviewer #1: Yes

Reviewer #2: Yes

Reviewer #3: Yes

4. Have the authors made all data underlying the findings in their manuscript fully available?

Reviewer #1: Yes

Reviewer #2: Yes

Reviewer #3: Yes

5. Is the manuscript presented in an intelligible fashion and written in standard English?

Reviewer #1: Yes

Reviewer #2: Yes

Reviewer #3: Yes

6. Review Comments to the Author

Reviewer #1: The authors have revised the manuscript very well and now acceptable for publication. However, I have a few minor comments that need to be looked after for future.

1. Do the correlation of pI and Molecular weight has any impact on structural aspects of the protein?

2. Can these Molecular weight and pI play role in structure of the protein?

Reviewer #2: (No Response)

Reviewer #3: The authors have addressed all the comments. However there are a very few grammatical errors should be addressed.

I recommend this article for publication.

7. PLOS authors have the option to publish the peer review history of their article (what does this mean?). If published, this will include your full peer review and any attached files.

Reviewer #1: No

Reviewer #2: **Yes: **Vinod Kumar Yata

Reviewer #3: No

---

## [Author Response · Author response to Decision Letter 1]

10 Sep 2022

Dear Editor,

Greetings

Please find the attached letter to find the response to reviewer commnts.

Regards

Dr. Tapan

---

## [Editor Report · Decision Letter 2]

12 Sep 2022

Virtual 2D Map of Cyanobacterial Proteomes

PONE-D-22-12929R2

Dear Dr. %Mohanta%,

We’re pleased to inform you that your manuscript has been judged scientifically suitable for publication and will be formally accepted for publication once it meets all outstanding technical requirements.

Kind regards,

Arabinda Ghosh

Academic Editor

PLOS ONE

---

## [Editor Report · Acceptance letter]

23 Sep 2022

PONE-D-22-12929R2 

Virtual 2D Map of Cyanobacterial Proteomes 

Dear Dr. Mohanta:

I'm pleased to inform you that your manuscript has been deemed suitable for publication in PLOS ONE. Congratulations! Your manuscript is now with our production department. 

Kind regards, 

on behalf of

Dr. Arabinda Ghosh 

Academic Editor

PLOS ONE